# Impetigo Animal Models: A Review of Their Feasibility and Clinical Utility for Therapeutic Appraisal of Investigational Drug Candidates

**DOI:** 10.3390/antibiotics9100694

**Published:** 2020-10-14

**Authors:** Solomon Abrha, Andrew Bartholomaeus, Wubshet Tesfaye, Jackson Thomas

**Affiliations:** 1Faculty of Health, University of Canberra, Bruce, Australian Capital Territory 2617, Canberra, Australia; Solomon.Bezabh@canberra.edu.au (S.A.); a.bartholomaeus@uq.edu.au (A.B.); Wubshet.Tesfaye@canberra.edu.au (W.T.); 2Department of Pharmaceutics, School of Pharmacy, College of Health Sciences, Mekelle University, Mekelle 7000, Ethiopia; 3Daimantina Institute, University of Queensland, Wolloongabba, Brisbane 4102, Queensland, Australia

**Keywords:** animal models, hamsters, impetigo, in vivo evaluation, mice, review, skin infections, topical treatment

## Abstract

Impetigo (school sores), a superficial skin infection commonly seen in children, is caused by the gram-positive bacteria *Staphylococcus aureus* and/or *Streptococcus pyogenes*. Antibiotic treatments, often topical, are used as the first-line therapy for impetigo. The efficacy of potential new antimicrobial compounds is first tested in in vitro studies and, if effective, followed by in vivo studies using animal models and/or humans. Animal models are critical means for investigating potential therapeutics and characterizing their safety profile prior to human trials. Although several reviews of animal models for skin infections have been published, there is a lack of a comprehensive review of animal models simulating impetigo for the selection of therapeutic drug candidates. This review critically examines the existing animal models for impetigo and their feasibility for testing the in vivo efficacy of topical treatments for impetigo and other superficial bacterial skin infections.

## 1. Introduction

Impetigo (school sores) is one of the eight dermatologic conditions listed in the 50 most common causes of infection globally [1], and is the only one of those eight with significant potential for life-threatening complications such as post-streptococcal glomerulonephritis and rheumatic heart diseases [2]. Over 162 million children suffer from impetigo globally at any given time, with a disproportionately high prevalence reported in Indigenous Australian children [3]. Impetigo is epidermal infection caused by the gram-positive bacteria *Staphylococcus aureus* and/or *Streptococcus pyogenes* (group A beta-hemolytic streptococcus; GAS) [3,4]. It can occur either as a primary infection or secondary to other underlying skin conditions which disrupt the skin barrier and allow the entry of pathogenic bacteria such as atopic dermatitis, eczema, or scabies [5,6]. Impetigo presents in two forms—bullous and non-bullous [4]. Bullous impetigo is generally caused by *S. aureus* and clinically presents with large fragile, flaccid, fluid-filled bullae (up to 2 cm in diameter) that less readily rupture into thin, brown crusts [4,7]. Non-bullous or crusted impetigo is the most common form of impetigo caused by both *S. aureus* and GAS [7]. It begins with a vesicle located on an erythematous base, which is easily ruptured, resulting in superficial ulceration covered with purulent discharge that then dries as an adhering yellow crust. Clinically, non-bullous impetigo caused by GAS is usually indistinguishable from non-bullous impetigo caused by *S. aureus* [8]. The clinical appearances of impetigo forms may be dependent on the type of disease-causing bacteria or etiologic agent; however, the histopathology of both forms is suggested as similar and is mainly denoted by the formation of intraepidermal pustules [9].

Antibiotics, often topical, are used as the first-line treatment for impetigo for faster symptom resolution [4,6,10,11]. However, as for other infections caused by *S. aureus* and *S. pyogenes*, antimicrobial resistance (AMR) poses a serious challenge in impetigo treatment and the rapid emergence and spread of multidrug-resistant bacterial strains (methicillin resistant *S. aureus* (MRSA) and macrolide-resistant *S. pyogenes*) is a global concern, limiting the effectiveness of existing treatment options for infections caused by these strains [12,13,14,15,16]. Given the seriousness of AMR, the World Health Organization (WHO) and the research community have recently highlighted the critical need for research of resistance patterns of pathogens to the existing antibiotics and the development of new and alternative antimicrobial agents with a number of new and existing candidates under investigation [10,17,18,19,20]. The antimicrobial activity of potential new compounds is first tested in vitro and, if effective, followed by study in animal models and clinical studies [21,22].

Appropriate animal models provide a highly controlled and cost-effective technique to predict efficacy in humans and refine treatment regimens and delivery vehicles bridging the gap between the in vitro drug discovery and clinical studies [22,23,24,25,26]. Animal models allow investigators to study the efficacy and safety of antimicrobial agents against specific pathogens in the presence of various host factors, metabolic processes, and anti-infective host defense mechanisms in the skin [22,23,27]. The use of animal models for the evaluation of new antimicrobial agents or new skin therapies is a long-established practice and has been acknowledged as a prerequisite and integral part of preclinical studies in drug development [21,28,29,30]. In addition to efficacy data, animal models provide other preliminary information, including safety, pharmacokinetic, and pharmacodynamic profiles of the potential antimicrobial agents necessary for the design and conduct of subsequent clinical studies [21,28,29,31,32].

To obtain the greatest scientific value from models, the choice of model must be optimized for the particular disease of interest and should react to the disease or its respective treatment in a way that resembles human physiology [26,33]. When selecting an optimum animal model, consideration should be given to the clinical relevance, reliability and reproducibility of the data generated, the technical complexity of conducting and interpreting the study, and the ethical acceptability of the animal use in the study [33,34].

For animal models aimed at screening and evaluating potential antibacterial agents for human use, the models should be able to establish the infections of the target human pathogens [34,35]. The models should allow for the observation of measurable study outcomes which are reproducible and readily controlled [34,35]. The study parameters should also be sensitive enough to allow detection and quantification of both bacteriostatic and bactericidal effects of the agents; enable clinically active drugs to demonstrate their efficacy; and provide results that show a high degree of clinical predictivity [34,35].

Although the results generated from animal models can and do provide a degree of clinical predictivity and are the best preliminary screening methods to select new antimicrobial agents or therapies for clinical trials, they cannot be directly translated to humans and the final judgment of an antimicrobial agent’s efficacy is ultimately dependent on the results of controlled clinical trials [21,22,26,28,29]. Over the years, several models of skin infection have been developed with considerable success in reproducing the pathogenesis of *S. aureus* and GAS diseases and are currently being used to study skin and soft tissue infections (SSTI) caused by these bacteria [30,34,36,37,38]. The experimental details of these models have been extensively summarized in a number of reviews [30,36,37,39]. However, the same cannot be said for impetigo at present, and there is a lack of a comprehensive review on the animal models investigated for impetigo. Here, we critically review the animal models investigated to mimic clinical impetigo and discuss their methodology, strengths, limitations, and the rationale for their application in evaluating impetigo drug candidates.

A summary of the various animal models is presented in Table 1 and Figure 1, and the strengths and limitations of the models are also summarized in Table 2.

### 1.1. Syrian Hamster Impetigo Model

This model was established in 1970 by Dajani and his co-workers using experimental 6–8-week-old golden Syrian hamsters [9,40]. Subsequent studies have used this model to investigate the effect of various therapeutic regimens [41], the interaction between *S. aureus* and GAS strains [42], and the cellular responses following the infections [43]. In all studies, the authors have followed a similar procedure to establish the infection. The back of the hamster was carefully shaved and intradermal injections of 0.1 mL of fresh cultures containing 1.2 × 10^8^ Colony Forming Units (CFU)/mL of test organisms (*S. aureus* or GAS) were given at four to six sites at least 3 cm apart, with a 27-gauge needle fitted to a tuberculin syringe. After 24 h of the inoculation, various treatment regimens such as scrubbing with hexachlorophene soap (once daily, no report on dose and duration of treatment), gentamicin ointment (twice daily, no report on dose and duration of treatment), bacitracin ointment (twice daily, no report on dose and duration of treatment), benzathine penicillin G injection (30,000 U, one-time IM injection), or procaine penicillin injection (3000 U daily, for 7 days) were initiated for only GAS-infected hamsters and compared with a control (no treatment) group. Then, the infected hamsters were carefully observed daily for about 15 days for the gross clinical appearance of the lesions and the effect of therapy on the persistence of organisms in the lesions. The appearances of the lesions were examined photographically, and the effect of the treatments was investigated by swabbing the lesions for the presence of residual organisms.

In their results, the authors observed a formation of papular, erythematous areas, 4–6 h after injection, which progressed into vesicles within 24 h. The vesicular lesions then ruptured and subsequently became crusted within 2–3 days and remained in the crusted stage for 4–6 days. Histopathology demonstrated inflammatory cells invading muscle, adipose tissue, and the upper part of the dermis as well as the epidermis. Based on these observations, the authors concluded that the lesions developed in this model simulate the human disease in gross appearance, disease progression, and the histopathology of the infection process. Comparing features of the lesions produced by the test organisms, the authors noted that *Staphylococcal* and *Streptococcal* lesions were similar except that no vesicular stage was observed, and cellulitis was more pronounced with *Staphylococcal* infection.

The authors also found that the healing process for systemic treatments (mean duration: 4–5 days) was relatively faster than for topical treatments (mean duration: 7–8 days) compared with control groups (6 days), with complete eradication of the organism observed for only systemic treatments after 15 days of treatment, validating the applicability of this model for the therapeutic evaluation of drugs. 

Even though the model succeeded in producing a human impetigo-like infection, the mode of bacterial inoculation (deep intradermal injection) is unlikely to represent the natural progression of impetigo development in humans. Human impetigo initially develops as a small vesicle after minor injuries or cuts on the skin, such as abrasions and skin scratches resulting from eczematous lesions or insect bites, rapidly progressing into pustular lesion, and subsequently evolving into an erosion covered by a thick crust [8,60,61,62]. Other authors [45,46,49,50,63] also suggest that the application of topical inoculation of organisms to the surface of slightly damaged or traumatized skin is likely to more closely resemble the natural route of transmission for GAS and *S. aureus*. Minor skin traumas perforate the stratum corneum, causing a temporary breach in the integrity of the skin barrier system that enables the inoculated bacteria to permeate or penetrate the epidermis with little hindrance [62,64]. In addition, the inoculum size of the test organisms (1.2 × 10^7^ CFU given at four sites) used to establish infection in the hamster model studies was too large in comparison to natural conditions. In human subjects, inoculum sizes of 10^6^ cells (*S. aureus*) and 10^4^ cells (GAS) are reportedly sufficient to produce severe *Staphylococcus aureus* skin infections [64].

The experimental impetigo produced in this model was also associated with a subepidermal abscess, a sign of the formation of dermal (e.g., cellulitis) and deep tissue infections, which is usually treated with systemic antibiotics in clinical settings [62,65]. This seems to negate the claim of establishing an impetigo-like infection (which is usually confined to the epidermis) [45,65]. The large inoculum delivered subcutaneously resulting in abscess formation creates the risk that results from the model will be biased toward systemically administered antimicrobials and lead to rejection of potentially clinically useful topical treatments [49].

Although the description of the sampling procedure in the studies lacked detail, swabbing and culturing of the samples were performed. This approach may confirm the presence of the test organisms, but it is unlikely to give a reliable estimate of the total count of organisms present in the lesion sites. Recent similar studies [52,53,54] suggest the use of quantification of viable lesion bacteria count through removing skin samples from the lesion sites with subsequent homogenization and culturing of the supernatant. This helps to quantify organisms present at both the surface and superficial tissue and provides a measure of the total count of bacteria contributing to the infection [52,53,54].

The hamster model was designed almost four decades ago, and currently little data on the response of this model to currently marketed topical antibiotics is available [44]. In addition, the use of hamsters for experimental purposes in the investigation of immune responses to skin infections has now been diminished due to a substantial difference between the human and hamster immune systems [66].

In sum, this model appears to have limited value in evaluating in vivo efficacy of potential topical antibacterial agents targeting impetigo and other SSTIs.

### 1.2. Mouse Skin Abrasion Impetigo Model

There has been a preference for mice (*Mus musculus*) as a model for SSTIs in recent years and mice are currently the most widely employed animal to model skin and other infectious diseases [24,36,67], not only because of their relative genomic similarity with humans, but also because of their availability, ease of handling, high reproductive rates, and relatively low cost [25,36,68]. Due to their ability to mimic the clinical presentations of *S. aureus* and GAS infections observed in humans, mouse models are most extensively used for studying the pathogenesis of infections caused by *S. aureus* and *S. pyogenes* (GAS) [24,30,36,59]. They are also capable of discriminating the clinical presentation of infections caused by different strains of disease-causing bacteria [69].

Recognizing the limitations of the hamster model, Abe et al. (1992) explored a mouse model for human skin infections [45]. In this study, female five-week-old mice (~20 g, ddY—Deutschland, Denken, and Yoken type) were treated with cyclophosphamide (Cy; 2 mg/mouse, intraperitoneal injection for 5 days), a potent immunosuppressive agent, to elicit leukocytopenia. The intent of the immunosuppression was to facilitate clear observation of the interaction of *S. aureus* and the epidermal cells during infection, as white blood cells (particularly neutrophil) infiltration during infection can obscure this interaction. Non-Cy treated mice were also used as a comparative control. After Cy-treatment, the back of the mice was carefully shaved using razor blades and the exposed skin was then slightly abraded three times using sandpaper. Fifty µL of *S. aureus* suspension (1.4 × 10^7^ CFU/mL or 7 × 10^4^ CFU) was then applied topically on each of the abraded areas, and the inoculated sites were occluded with sterile plastic plasters and sealed with vinyl adhesive tape. Subsequently, skin samples (1.5 × 1.0 cm^2^, n = 3) were excised from each mouse at different times (0.25–48 h) after the inoculation. The skin samples were washed with saline solution, cut into small pieces, and homogenized in saline solution (2 mL) in sterile mortars and aliquots (0.1 mL) cultured on appropriate *Staphylococcus* culture media. After 24 h of incubation, CFUs were counted and the resulting counts were converted to CFU/g skin sample. Additionally, skin samples were examined histopathologically and electron microscopically to assess the disease progression across the skin layers.

The study revealed the presence of bacteria in the horny skin layer (stratum corneum) of the Cy-treated mice at 0.5 h after inoculation, and the formation of subcorneal (immediately below stratum corneum) bullae at 1 h, with subsequent bacterial cluster formation inside the bullae at 3 h. After 6–12 h following inoculation, intraepidermal bullae containing many bacteria were developed, became enlarged, and spread to the upper dermis accompanied by epidermal necrosis within a day (at 24 h of the inoculation). Two days (at 48 h) after inoculation, bacteria further invaded into some surrounding hair follicles with a slight leukocyte infiltration. The non-Cy-treated mice showed the same outcome except for the abundant leukocytes and bacteria at 6 and 48 h; however, the authors suggested that further experimental models without the use of Cy should be explored. 

In summary, the authors stated that the model has the potential to produce impetigo lesions that closely resemble the human blister lesions in terms of intraepidermal pustules formation and pathology findings. They also emphasized the importance of occlusion by an impermeable plastic film for the development of human-like *S. aureus*-induced blisters. Similar studies aimed at investigating the pathogenesis of *S. aureus* and GAS using in vitro (human skin explant culture) [70], in vivo (mouse model) [46], and human subjects [62,63,64,71] also reported that occlusion is necessary for the development of experimental skin infections. In two studies involving human subjects [64,71], bacterial inoculation was made on the intact skin, and occlusion was used to enhance the skin hydration, because moisture is suggested as the main factor promoting the growth of bacteria on the skin surface. The resulting moist and warm environment provides ideal conditions for the rapid growth of bacteria and subsequent development of the infection model [71].

The mouse skin abrasion model is designed specifically to simulate impetigo triggered by *S. aureus* sp. and further studies are required to confirm if the model is flexible enough to accommodate additional pathogens (e.g., *S. pyrogens*). Producing skin abrasion using sandpaper has also been viewed as a difficult procedure to monitor and control [46]. Moreover, we could not identify additional studies in the literature that used this model for the evaluation of topical antibiotic agents for SSTIs. Thus, it is difficult to validate the reproducibility and reliability of this model.

### 1.3. Humanized Mouse Model

Scaramuzzino et al. (2000) developed a humanized mouse (hu-mouse) model for impetigo by grafting human epidermal tissue from neonatal foreskin onto the back of 4–6-week-old female severe combined immunodeficiency disease (SCID) mice (mice that lack functional B and T lymphocytes), followed by infecting the graft with GAS four weeks after surgery [46]. To test the effect of compromised/damaged skin, the grafts were superficially damaged by one of three methods: a series of gentle cuts with a scalpel blade, gentle rubbing with sandpaper, or tape stripping (stripping the graft ten times in sequence) using fresh duct tape. An inoculum of GAS (50 µL of 10^3^ CFU/mL) on a gauze pad was applied to the damaged area of skin under a circular bandage and occlusive dressing. A week after the inoculation, the mice were sacrificed, and their human skin grafts (1 × 1.5 cm) were removed for histopathology examination and determination of the bacterial count.

The study confirmed that the formation of an impetigo-like lesion characterized by erosion of the stratum corneum, and a typical murine innate immune system response (infiltration polymorphonuclear leukocytes), were observed. The typical honey-colored fibrin crust typical of non-bullous impetigo in humans was not observed, however. Only small inoculums of as low as 50 CFU of GAS were required to produce the impetigo-like lesions. In addition, the study confirmed that damaged skin and subsequent occlusion of the inoculated skin area were requirements for the induction of severe infection by virulent GAS. Other studies [45,62] have also recommended the requirement of breaching the integrity of the skin through minor superficial damage to elicit the *Streptococcal* infection.

The authors noted that tape stripping could not reliably remove the stratum corneum and it occasionally led to the separation of large stretches of the epidermal layer from the underlying dermis; whereas the use of sandpaper treatment was found difficult to monitor and control. These authors suggested that a gentle crosswise scalpel cut method was the most reproducible method tested to achieve superficial skin damage.

Given the use of grafted human skin followed by the topical application of low bacterial inoculum to superficially damaged grafted skin, this model potentially provides the closest simulation of the clinical pathology of impetigo [30,59]. This model has been employed in studies [47,48] seeking a better understanding of the pathogenic mechanisms associated with a GAS skin infection and has been suggested as an important model that could pave the way for the development of a vaccine for streptococcal-caused impetigo [59].

The hu-mouse model is associated with a number of important limitations. It is technically complex, requiring a source of human tissue and surgical and post-surgical animal nursing skills to successfully achieve the grafts [30,72]. The absence of an adaptive immune response in this mouse line prevented the rejection of the tissue grafts, but also removed a potentially relevant immune mechanism in the pathogenesis of impetigo [46]. The use of an immunocompromised SCID mouse is unlikely to closely simulate the physiological conditions encountered by the host under natural conditions, as the infection proceeds in the absence of adaptive immunity in this type of mouse [30]. The model is also likely limited by the requirement of a large number of mice and a huge effort to perfect the infection model [30]. Moreover, this model has been applied only to GAS, and its applicability for *S. aureus*-driven impetigo is unexplored. No relevant studies using this model for therapeutic evaluation of topical antibiotics for impetigo were located in the literature.

A reliable and reproducible mouse skin infection model with established pathogenesis of *S. aureus* and *S. pyogenes* is a prerequisite for further studies evaluating the therapeutic response of agents [63]. Our review identified a limited number of existing models developed specifically for impetigo and only the humanized mouse model seems to produce a human impetigo-like infection. In addition, the identified models have a number of limitations that restrict their feasibility in evaluating the clinical efficacy of topical antibacterial agents. A number of other models have been developed to investigate superficial skin infections more broadly [34,49] and for assessing the clinical efficacy of both marketed and investigational topical antibiotics (such as fusidic acid, mupirocin, retapamulin, and ozenoxacin) for impetigo treatment, and these are discussed below.

### 1.4. Mouse Skin Tape-Stripping Model

Kugelberg et al. (2005) established this model using 6–8-week-old female BALB/c mice, a readily available albino inbred mouse strain [49]. A reproducible degree of dermal damage was produced using tape stripping 7–10 times with an elastic adhesive bandage, standardized by measuring the trans epidermal water loss (TEWL) using a proprietary TEWL probe to yield a TEWL of approximately 70 g/(m^2^∙h). Following this procedure, the skin became visibly damaged, characterized by reddening and glistening but no regular bleeding. Then, an inoculum of 10^7^ cells of *S. aureus* or *S. pyogenes* in a 5-µL droplet was placed on the stripped skin to initiate the infection. Four hours post infection, the mice were treated with 2% fusidic acid ointment twice daily (AM and PM, with an 8- hour interval) for a period of 4 days. Eighteen hours after the last topical treatment, the mice were killed, and the wounds (about 2 cm^2^) were excised, processed, and cultured to determine the number of viable bacteria per unit area of skin. In order to investigate the reproducibility of the infection with *S. aureus* and *S. pyogenes*, three independent experiments that included untreated and placebo-treated groups were performed.

This study concluded that the model is simple, reproducible, and useful to simulate localized skin infections caused by *S. aureus* or *S. pyogenes*. Topical treatment with fusidic acid significantly reduced the number of viable counts of *S. aureus* and *S. pyogenes* after the four-day treatment (*p* < 0.001), showing that this model can also be used for therapeutic evaluation of topical treatments. However, the authors stated that this model did not simulate the histology of impetigo. In human impetigo involving *S. aureus*, the first step for the bacterial skin invasion involves adhering to the outer layer of the epidermis (stratum corneum), followed by disruption of the epithelial barriers of the inner layers of the epidermis comprising cell-adhesion structures such as desmosomes and adherence junctions [73]. This process separates the epidermis just below the stratum granulosum and large intraepidermal pustules, which may contain bacteria, and these are described as typical histologic characteristics of impetigo [45,50,74]. Besides, in the case of impetigo, the upper dermis also contains an epidermal inflammatory infiltrate of neutrophils and lymphocytes which are reported to be crucial for the staphylococcal percutaneous invasion [50].

In this model [49], there was no intraepidermal or sub-corneal pustules nor the migration of neutrophils, which is inconsistent with the picture for human impetigo. Similar studies [50,51] that used this model and followed the same procedure have reported contrasting results. One study [50] claimed the formation of intraepidermal pustules resembling human impetigo through epicutaneous inoculation of *S. aureus* on the inner pinna of the mice. In contrast, other studies [51,75] demonstrated the formation of more of a cutaneous infection rather than human impetigo-like infection following a similar inoculation procedure and concluded that, in this mouse model, the test organisms are capable of rapidly penetrating epidermal layers and disseminate into dermal and cutaneous tissues. Similar results were also reported by other investigators [76,77] employing this model to evaluate a new topical antibiotic (HT61) for methicillin-sensitive S. *aureus* (MSSA) and MRSA infections. These studies showed that tape-stripping the mouse skin using autoclave tape ten times in succession damaged the skin by removing the top dermal layers, which became red and shiny but without observable bleeding—meaning the removal of most of the epidermal layer of the mouse skin. From these inconsistent findings, it is understood that the mouse skin tape-stripping is unlikely to be a reliable and reproducible model for impetigo.

### 1.5. Mouse Suture-Superficial Skin Infection Model 

This model was originally investigated in two successive studies with the aims of characterizing the infection, evaluating its response to therapy through testing various topical antimicrobial formulations, and investigating the correlation of the in vitro antimicrobial activity of the tested formulations with their in vivo efficacy [34,35]. Accordingly, evaluation of in vitro antibacterial activity (minimum inhibitory concentration and minimum bactericidal concentration) of the topical agents was performed following the broth twofold dilution method. For the in vivo study, an infection model was developed using female CF-1, 18 to 20 g mice (n = 10 in each group). The study further explored the correlation of in vitro with in vivo study data.

For the in vivo study, mice were anesthetized; the dorsal area closely shaved; and on the day of infection, superficial surgical wounds were produced by making a longitudinal midline incision of 2.3 cm in length. The wounds were then infected either by direct seeding or by the insertion of a 5 cm length suture, which had been previously infected with a controlled inoculum (10^3^–10^5^ cells) of *S. aureus* or *P. aeruginosa*, using a surgical needle. To maintain its position, the ends of the suture were then secured with rubber cement. 

In this model, two treatment regimens (immediate and delayed) were employed to assess the efficacy of the test formulations (gentamicin cream, neomycin-gramicidin-nystatin-triamcinolone acetonide cream, nitrofurazone cream, polymyxin B-bacitracin-neomycin ointment, and triclobisonium chloride ointment). In the immediate-treatment regimen, each wound was treated topically with 0.4 g of formulation 15 min and 6 h after the insertion of the suture; whereas the delayed treatment regimen was administered 24 and 30 h after suture insertion. For both regimens, the wounds were excised and cultured 18 h after the final treatment.

Quantitation of the viable bacteria in the wounds was performed before and after the treatments. The total pre-measured amount of viable bacteria counts loaded in the suture was used as a baseline, or pre-treatment viable count; whereas for post-treatment viable count determination, both surface rinse and biopsy (skin sample or tissue) homogenization techniques were performed in the study. The surface rinse technique was performed by tightly placing a sterile test tube containing distilled water in an inverted position on the back of a euthanized mouse in such a way that more than 90% of the wound was enclosed by the mouth of the tube. Then, the mouse and the test tube were held together, with the tube upright, and shaken vigorously for 25 times to remove the infecting organisms from the surface of the wound. The samples were then processed for determination of CFU. In the case of the biopsy homogenizing method, biopsy samples from infected wounds were taken, homogenized, diluted, and plated on the appropriate selective agar media for determination of CFU.

The results of these successive studies [34,35] indicated that clinical signs of infection like oedema, erythema, and suppuration were observed around the margins of the lesions, representing the secondary skin infections like impetigo which occur following skin damages. The developed infection persisted for 1 to 2 weeks and was found to be quite susceptible to both immediate and delayed regimes of topical treatments with efficacy consistent with in vitro activity against the tested organisms, demonstrating a reasonable correlation of the in vitro activity of the topical agents with their in vivo efficacy in the animal model.

Unlike the suture method, the directed seeding infection method resulted in greater variation in viable wound bacteria counts—this was chiefly attributed to inoculum runoff from the application site. However, only an inoculum size of 10^3^–10^5^ cells of the test organisms was required to elicit the desired experimental infection using the suture method. This inoculum size is even lower than the inoculum dose (10^6^ cells) used to initiate *S. aureus* infection in human subjects [64,78]. The simplicity of the surgical procedure used in this model and a low inoculum dose needed to elicit the infection, are important advantages of this model. The suture method for eliciting bacterial skin infections (*S. aureus* or *S. pyogenes*) has also been used by other studies using mice [44,52,53,55,57,58,79], guinea pigs [56], and humans [80], reflecting some acceptance of the use of sutures to introduce a bacterial inoculum as a means to simulate superficial bacterial skin infections in humans [78].

For quantification of the viable bacteria count in skin, although the surface rinse method resulted in a lower count per unit area, it nonetheless correlated well with the dynamics of the total viable wound count obtained from the biopsy method. The skin surface rinse approach was faster, simpler and less labor intensive, and the authors therefore recommended this approach for screening type studies. For studies aiming to investigate the therapeutic efficacy of topical formulations, the biopsy homogenizing, as extensively used by similar studies [44,55,79], seems more appropriate, as it quantifies both the surface and superficial tissue-associated bacteria. Although expensive, technically complex, and relying heavily on a large number of approximations compared with conventional methods [81], using an optical in vivo bioluminescence imaging system could potentially ease the determination of in vivo bacterial burdens without excising the skin samples [82,83]. This technique uses bioluminescent bacteria strains, which are placed in the full thickness incision in the dermis of the animal skin, and capture the bioluminescent signals released by the bacteria with a camera to determine the bacterial burden [82].

Even though the mouse suture-superficial skin infection model was original established as a skin wound infection, it has been extensively advanced and optimized by researchers over the past few years to study the superficial skin infections like impetigo. In fact, this model is arguably the most widely used mouse model for the clinical evaluation of currently marketed topical antibiotics for the treatment of impetigo and SSTIs, such as retapamulin [54], fusidic acid [44], mupirocin [44,56], amoxicillin or amoxicillin-clavulanic acid [55], Gemifloxacin [57], as well as other new impetigo and SSTIs agents in the pipeline [52,53,58]. This could be because of the ability of the model to simulate topical skin infections like impetigo that are caused by either *S. aureus* alone, GAS alone, or both in combination [55], and this allows the researchers to study the effect of a broad spectrum of topical antibiotics against both bacteria. Moreover, the efficacy results obtained from this model have been shown to positively correlate with the efficacy results obtained from clinical trials in humans [44,52,54]. For instance, ozenoxacin 1% cream [84,85], retapamulin 1% ointment [86], mupirocin 2% ointment [87], and mupirocin 2% cream [88], studied using this model, have subsequently shown good efficacy in clinical trials, indicating the potential of this model in predicting the efficacy of topical antibiotics in human [54]. Its capability to respond to various antibiotics from a wide range of dosage forms, including cream, ointment, gel, solutions, and powder, makes this model uniquely suitable for the evaluation and comparison of the efficacy of topical and oral antibiotics [44,52,54,55,56]. Although most of the studies using this model followed the original procedure, which is the incision of the skin with a scalpel and placing the infected suture in the wound, it is worth mentioning the slight modification of this model reported in other studies [44,52,55], which performed insertion of the infected suture under the skin using a surgical needle followed by making a superficial incision using a scalpel along the length of the suture without reaching the panniculus carnosus. Piercing of the skin using a surgical needle followed by superficial skin incision could represent the natural progression of impetigo in humans, i.e., scratching an insect bite, minor injuries, or cuts on the skin, which subsequently lead to the entry of impetigo-causing bacteria. Two studies [44,56] also used an adhesive temporary skin closure to ensure the wound remained closed until treatment initiation.

In summary, the mouse suture-superficial skin infection model appears to be the most appropriate and practically viable in vivo method for evaluating the antimicrobial efficacy of topical skin formulations where the primary focus is impetigo.

## 2. Concluding Remarks

Relevant, predictive skin infection models are required for the preclinical, in vivo investigation of the safety and efficacy of new topical antimicrobial agents. In this regard, there is a limited range of in vivo models that can reliably simulate impetigo and the efficacy of prospective treatment. Given impetigo usually occurs as a mixed bacterial infection caused by *S. aureus* and GAS species, none of the models investigated for impetigo seem to simulate impetigo caused by co-infection with both bacteria. Almost all the models used topical inoculation of the bacteria to the surface of slightly damaged skin and this delivery method seems to correspond well with the route of transmission encountered during natural infection resulting in impetigo. In addition, the majority of the models recommend occlusion of the topically inoculated skin surface to maintain the moisture required for the growth of bacteria, particularly for the GAS species. It is also worth noting that all mouse models preferred female mice as opposed to their male counterparts. This seems to relate to the lesser likelihood of female mice involved in cage fights for dominance, resulting in fight wounds and related infections, which could considerably interfere with the experimental observations.

Given the strengths and limitations of models reviewed, when planning for the in vivo efficacy evaluation of potential topical antibacterial therapies, key considerations beyond their practical feasibility in a given laboratory should include: (1) the establishment of reproducible, controllable infections with measurable endpoints of efficacy, (2) reliable and simple quantification of pre- and post-treatment viable bacteria counts, and (3) the correlation of in vivo results with clinical effectiveness in humans.

While the humanized mouse model appears to be a promising infection model, as it can reproduce many features of human impetigo, it is limited by the requirements for human tissue and the need for experienced and surgically skilled personnel. So far it has only been studied for the pathogenesis of GAS and it lacks evidence to support its utility for the evaluation of topical antibiotics.

For practical simplicity and reproducibility, the mouse suture superficial skin infection model has many advantages and has been widely used to study the preclinical efficacy of marketed topical impetigo treatments. This method has been shown to be suitable to study the pathogenesis of impetigo-causing bacteria and appears to be the model of choice for researchers and pharmaceutical companies for both screening and therapeutic evaluation of investigational drugs for the topical treatment of impetigo.

## Figures and Tables

**Figure 1 antibiotics-09-00694-f001:**
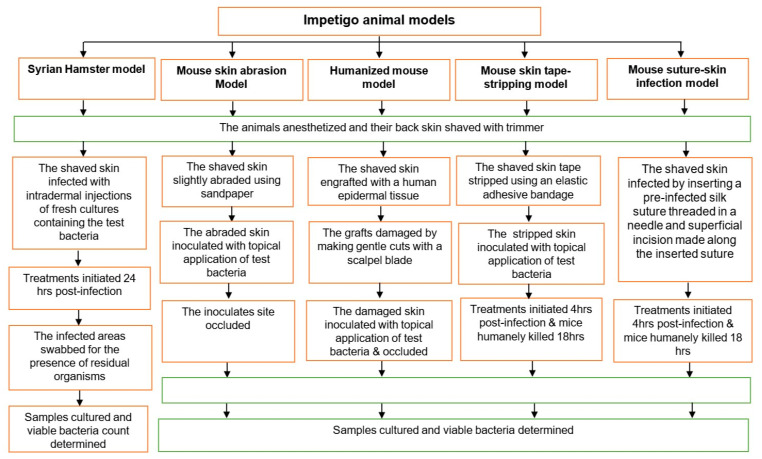
Summary flow diagram of the rodent models investigated for impetigo.

**Table 1 antibiotics-09-00694-t001:** A summary of rodent models investigated for impetigo.

Model	Bacteria Tested	Host Animal	Anaesthetic Agent	Inoculation Route	Inoculum Dose	Time for Infection Occurrence	Antimicrobial Agents Tested	Sampling Method
Hamster impetigo model [9,40,41,42,43,44]	*S. aureus* or *S. pyogenes*	Hamster (Golden Syrian type, 6–8 weeks, 80–120 g, n = 4–75)	Isoflurane (3%, inhalation)	Intradermal injections	1.2 × 10^7^ CFU	24 h post-inoculation	Gentamicin ointment, Bacitracin ointment, Benzathine penicillin G injection, Procaine penicillin injection	Swabbing the lesion surface
Mouse skin abrasion impetigo model [45]	*S. aureus*	Mouse (ddY type, female, 5 weeks old, ~20 g, n = 5)	Not reported	Topical/epicutaneous inoculation to slightly damaged skin by sandpaper	7 × 10^4^ CFU	24 h post-inoculation	No antimicrobial agent used	Biopsy of infected skin area
Humanized mouse impetigo Model [46,47,48]	*S. pyogenes*	Mouse (SCID type, female, 4–6-week-old, n = no report)	Ketamine-Xylazine (Intraperitoneal injection)	Topical/epicutaneous inoculation to slightly damaged skin by sandpaper, scalpel, or tape stripping	50 CFU	1 week post-inoculation	No antimicrobial agent used	Biopsy of infected skin area
Mouse skin tape-stripping model [49,50,51]	*S. aureus* or *S. pyogenes*	Mouse (BALB/c type, female, 6–8-weeks-old, n = no report)	1:1:2 *v/v* mixture of hypnorm (fentanyl, fluanisone), dormicum (midazolam) and distilled water, Intraperitoneal injection)	Topical/epicutaneous inoculation to slightly damaged skin by tape stripping	10^7^ cells	4 h post-inoculation	Fusidic acid ointment	Biopsy of infected skin area
Mouse suture-superficial skin infection model [34,35,44,52,53,54,55,56,57,58]	*S. aureus* and/or *S. pyogenes*	Mouse (CF-1, CD1, and MF1 type, female and male, 18–20 g, n = 10–50)	Sodium pentobarbital (30 mg/kg, Intraperitoneal injection) Or Diazepam plus fentanyl Fluanisone (1.25 mg/kg plus 0.5 mL/kg Intramuscular injection)	Topical/epicutaneous inoculation by insertion of an infected suture	10^3^–10^5^ cells	6 h post-inoculation	Gentamicin cream, Polymyxin B-bacitracin-neomycin ointment, Retapamulin ointment, Fusidic acid cream, Muprocin ointment and cream, Ozenoxacin cream, amoxicillin or amoxicillin-clavulanic acid oral, Gemifloxacin oral	Swabbing the lesion surface Or Biopsy of infected skin area

NB: The data included in the table are collated from multiple studies employing closely related methods/models where available.

**Table 2 antibiotics-09-00694-t002:** A summary of the strengths and limitations of the rodent models investigated for impetigo.

Models	General Strengths	General Limitations
Hamster impetigo model [9,40,41,42,43,44]	Widely studied and standardized method [9,40,41,42,43,44] Applicable to impetigo like infection from *S. aureus* or GAS species Relatively rapid development of impetigo-like infection, 2–6 days Simulates the human diseases in gross appearance, progression of the lesion (popular-vesicular-crusted-healing without scar), and histology Sample collection can be made through swabbing and culturing Rapid data generation—5 to 7 days The area of infection is usually on the dorsal side so that likely interference with the grooming behavior of the animal is limited	The mode of infection (intradermal inoculation) is different from that of human impetigo where infection initially develops following a trauma in the form of cuts, abrasions, and insect bites High bacteria inoculum is required to initiate the infection The experimental impetigo is associated with deep tissue infection (subdermal abscess) which potentially limits the clinical relevance of this model for drugs for topical or superficial skin infections Treatment starts 24 h post-infection as compared to other models [34] in which treatment is initiated 4–6 h after inoculation Hamsters have a different immune response as compared to humans
Mouse skin abrasion impetigo model [45]	Technically quick and simple model to perform Infection is initiated through topical inoculation, which is similar to human impetigo Experimental impetigo similar to bullous human impetigo Experimental duration 5–7 days The area of infection is usually on the dorsal side, hence less interference with the grooming behavior of the test animal	Designed for only *S. aureus* related impetigo Needs cyclophosphamide treatment to establish impetigo Could allow bacterial penetration through the skin surfaces if *Stratum corneum* is completely removed as a result of abrasion Sandpaper abrasion is difficult to standardize Utility for evaluation of treatments is not yet studied
Humanized mouse impetigo Model [46,47,48]	Produces an impetigo-like lesion [30,59] Topical inoculation, resembles natural process of human impetigo Can faithfully reproduce many features of human impetigo for *S. pyogenes* infection Sensitive to a relatively low inoculum to elicit the infection	Modeled only for GAS-caused impetigo No reports on the use of this model for treatment evaluation Complex model requiring human tissue and advanced surgical skills Needs humanized mice lacking functional B and T lymphocytes Time consuming, engrafted mice need 4 weeks of recovery before inoculation
Mouse skin tape-stripping model [49,50,51]	Technically quick and simple model to perform Developed to simulate skin infections caused by *S. aureus* or *S. pyogenes* Mode of infection mimics natural development of impetigo Can maintain the infection for 4 days after inoculation The tape-stripping is relatively painless and nonintrusive for the animals	Cannot accurately simulate impetigo as tape stripping could potentially remove the *Stratum corneum* and lead to separation of large stretches of the epidermal layer from the underlying dermis Could allow bacterial penetration into deeper skin structure No optimization of tape striping is done so it is difficult to regulate and monitor the tape stripping process
Mouse suture-superficial skin infection model [34,35,44,52,53,54,55,56,57,58]	Designed for both *S. aureus* and/or *S. pyogenes* Low dose is required to elicit the infection Well-established and widely used model to study efficacy of topical antibiotics marketed for impetigo (fusidic acid [44], mupirocin [44,56], and retapamulin [54]) Established for use as a screening tool for identifying and evaluating substances that may be employed as topical antibacterial agents either for systemic or local effects Widely used to select the dose and dosing regimen of a newer topical antibiotics for treating impetigo Confirmed responsiveness for various topical antimicrobial agents Can be used to conduct comparative study of oral, injectable, and/or topical antibiotics The infection could persist in the wounds for up to 7 days Results from this model are shown to correlate closely with efficacy in clinical trials with human subjects [44,52,54] The area of infection is usually dorsally located to hinder grooming or cleaning behavior of the animals	The throughput time per mouse (the total time taken to prepare a wound and inoculate it with bacteria) is about 20 min for this model as compared to 2 min for the tape-stripping model The time required to process each animal by this model could be labor intensive and time consuming when this model is used for antimicrobial screening purpose requiring many animals

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
