# Peer review of "Impetigo Animal Models: A Review of Their Feasibility and Clinical Utility for Therapeutic Appraisal of Investigational Drug Candidates"

_antibiotics, 2020, doi:10.3390/antibiotics9100694_

Round 1

Reviewer 1 Report

Overall a good paper, I think it addresses an unmet need and is a good review of the literature. I have a few minor suggestions:

  1. I recommend using numbered sections (i.e. 1. Introduction).
  2. The figures and tables look nice but I was wondering if that is the best placement in the paper for Figure 1/Table 1 as they are both placed before they are addressed.
  3. Content wise, the only thing I felt was missing was most of the models do a good job of addressing the similarity (or lack of) to impetigo, but this was not mentioned for mouse suture-superficial skin infection – so I would recommend adding it.
  4. The paper is well written but I just recommend one final grammar/spelling check as I saw only a few minor errors.

Author Response

Reviewer #1

GENERAL COMMENT

[Comment*]: Overall a good paper, I think it addresses an unmet need and is a good review of the literature. I have a few minor suggestions:

[#Response#]: We are thankful to reviewer #1 for the kind words and comments

Minor comments

  1. [Comment*]: I recommend using numbered sections (i.e. 1. Introduction).

[#Response#]: Thanks, and numbers are now used for topics and subtopics as recommended. (Please refer to Page #1–15)

  1. [Comment*]: The figures and tables look nice but I was wondering if that is the best placement in the paper for Figure 1/Table 1 as they are both placed before they are addressed.

[#Response#]: Thanks, and, given Figure 1 and Table 1 are the summary of the reviewed rodent models, we have now moved them to page 10–12. (Please refer to Page #10–12 and Line #405–408)

  1. [Comment*]: Content wise, the only thing I felt was missing was most of the models do a good job of addressing the similarity (or lack of) to impetigo, but this was not mentioned for mouse suture-superficial skin infection – so I would recommend adding it.

[#Response#]: Thanks for this comment. The two successive studies (McRipley RJ and Whitney RR, 1976a and 1976b), which originally established the mouse suture-superficial skin infection model, did not report the similarity (or lack of) to impetigo.

However, the authors reported the presence of clinical signs of infection like oedema, erythema, and suppuration around the margins of lesions, which seem to represent the secondary skin infections like impetigo which may occur following skin damages by accidental trauma, surgery, and burns. Nevertheless, this model has been extensively advanced and optimised by the researchers and industry over the past few years to study superficial skin infections like impetigo. Some studies using this model followed the original procedure, which is the incision of the skin using a scalpel and subsequently placing the infected suture in the wound. Whereas, other studies exploring topical antibiotics for impetigo, modified the procedure by performing insertion of the infected suture under the skin using a surgical needle followed by making a superficial incision using a scalpel along the length of the suture without reaching the panniculus. Suturing of the skin using a surgical needle followed by superficial skin incision collectively seem to mimic the impetigo pathology seen in humans, i.e. scratching an insect bite or intense itching from existing pathology like eczema, resulting in minor injuries or cuts on the skin, subsequently leading to the entry of impetigo causing bacteria.

Considering these, we have now included the following statements under this model to address this reviewer’s comment.

“The results of these successive studies [34, 35] indicated that clinical signs of infection like oedema, erythema and suppuration were observed around the margins of the infections, representing the secondary skin infections like impetigo which occur following skin damages.” (Please refer to Page #8 and Line #346–348)

“Even though the mouse suture-superficial skin infection model was original established as a skin wound infection, it has been extensively advanced and optimised by the researchers over the past few years to study superficial skin infections like impetigo.” (Please refer to Page #8 and Line #376–378)

“Piercing of the skin using a surgical needle followed by superficial skin incision could represent the natural progression of impetigo in humans, i.e. scratching an insect bite, minor injuries, or cuts on the skin, which subsequently lead to the entry of impetigo causing bacteria.” (Please refer to Page #8–9 and Line #397–400)

  1. The paper is well written but I just recommend one final grammar/spelling check as I saw only a few minor errors.

[#Response#]: Kindly noted and we have now checked the grammar and spelling as suggested.

Reviewer 2 Report

Manuscript ID: antibiotics-948596

The review entitled “Impetigo animal models: A review of their feasibility and clinical utility for therapeutic appraisal of investigational drug candidates” by Solomon Bezabh et al. describes an interesting review about the current in vivo models for impetigo skin disorder. The manuscript is well-written and well-organized. The reported literature goes deeply into the subject and gives an excellent overview of the topic. I think that this paper is suitable for the publication on this Journal, just after some corrections and suggestions that I would like to suggest to the author:

  • In the line 54, please correct “optionsfor” in “options for”.
  • Along with the manuscript, the authors sometimes used 6 hr, other times 6hr or 6hrs. Please use a unique format.
  • I noticed that in most of the reported mice models, the used mice were female. Do the authors know why female and not male? And if there are any advantages or disadvantages to this choice. I think it should be highlighted in the review.

Author Response

Reviewer #2

GENERAL COMMENT

[Comment*]: The review entitled “Impetigo animal models: A review of their feasibility and clinical utility for therapeutic appraisal of investigational drug candidates” by Solomon Bezabh et al. describes an interesting review about the current in vivo models for impetigo skin disorder. The manuscript is well-written and well-organized. The reported literature goes deeply into the subject and gives an excellent overview of the topic. I think that this paper is suitable for the publication on this Journal, just after some corrections and suggestions that I would like to suggest to the author:

[#Response#]: #]: We are thankful to reviewer #2 for such encouraging words and highly positive comments.

Minor comments

1.      [Comment*] In the line 54, please correct “optionsfor” in “options for”.

[#Response#]: Thanks, and it is now corrected as “options for”. (Please see Page #2 and Line #54)

2.       [Comment*] Along with the manuscript, the authors sometimes used 6 hr, other times 6hr or 6hrs. Please use a unique format.

[#Response#]: Thanks, and we have now consistently used 6hrs for all representations. (Please see Page #3 and Line #116; Page #7 and Line #332; Page #10–11 and Table 1, Column 7)

3.       [Comment*] I noticed that in most of the reported mice models, the used mice were female. Do the authors know why female and not male? And if there are any advantages or disadvantages to this choice. I think it should be highlighted in the review.

[#Response#]: Thanks for this insightful comment. The studies did not report the reason, advantages, or disadvantages for including only female mice. However, this could be due to the less aggressive nature of the female mice when they are placed in the same cage as opposed to their male counterparts. The male mice are likely to be involved in cage fights, and this usually create fight wounds and associated infections, which would interfere with the experimental observations.

We have now included the following statement in the conclusion part to address this comment.

“It is also worth noting that all mouse models preferred female mice as opposed to their male counterparts. This seems to relate with the lesser likelihood of female mice involving in cage fights for dominance, resulting in fight wounds and related infections, which could considerably interfere with the experimental observations.” (Please see Page #15 and Line #422–425)

Reviewer 3 Report

The manuscript by Abrha et al. entitled “Impetigo animal models: a review of their feasibility 2 and clinical utility for therapeutic appraisal of 3 investigational drug candidates” reviewed the impetigo animal models. Overall, this manuscript presents essential knowledge and technical progress of the models, which is of interest to the readers in this field. However, I have minor concerns:

1) In table 1, column title “time for infection occurence” is not accurate. It is the sampling time in most cases. In additon, the techniques of skin injury (abrasion, tape-stiping, suture, etc.) should be summaried in the table (in column inoculation route).

2) One literature by Guo et al. published in Antimicrobial Agents and Chemotherapy (2013;57/2: 855–863) should be included in this review, because it reported in vivo bioluminescence imaging to measure the bacterial burden.

Author Response

Reviewer #3

GENERAL COMMENT

[Comment*] The manuscript by Abrha et al. entitled “Impetigo animal models: a review of their feasibility 2 and clinical utility for therapeutic appraisal of 3 investigational drug candidates” reviewed the impetigo animal models. Overall, this manuscript presents essential knowledge and technical progress of the models, which is of interest to the readers in this field. However, I have minor concerns:

[#Response#]: We thank reviewer #3 for the positive feedback.

Minor comments

  1. [Comment*] 1) In table 1, column title “time for infection occurence” is not accurate. It is the sampling time in most cases. In additon, the techniques of skin injury (abrasion, tape-stiping, suture, etc.) should be summaried in the table (in column inoculation route).

[#Response#]: Thanks, we believe the times mentioned under the column ‘Time for infection occurrence’ refer to the duration between the inoculation of the bacteria and infection development in this case.

Regarding the techniques of skin injury, we have now added the respective skin damaging techniques to the respective models as per the recommendation. (Please see Page #10–11 and Table 1, Column 5)

  1. [Comment*] 2) One literature by Guo et al. published in Antimicrobial Agents and Chemotherapy (2013;57/2: 855–863) should be included in this review, because it reported in vivo bioluminescence imaging to measure the bacterial burden.

[#Response#]: Thanks for this information. We have now incorporated the following statement in the review to include the article in question (reference #83) and other related studies (references #81 and 82) in this revision.

“Although expensive, technically complex, and relies heavily on a large number of approximations compared with conventional methods [81], using an optical in vivo bioluminescence imaging system could potentially ease the determination of in vivo bacterial burdens without excising the skin samples [82, 83]. This technique uses bioluminescent bacteria strains, which are placed in the full thickness incision in the dermis of the animal skin, and capture the bioluminescent signals released by the bacteria with a camera to determine the bacterial burden [82].” (Please see Page #8 and Line #369–375)

Reviewer 4 Report

Title:  Impetigo animal models: A review of their feasibility and clinical utility for therapeutic appraisal of investigational drug candidates. 

General Comments: Very interesting review, enjoyed reading the manuscript.  It is well written and flows well.  I just have few minor comments for the authors for clarification and/or adjustment if they agree to do so. 

ABSTRACT

Line #17 - Line #18: The term “treatment” is repeated three times, consider rewarding the sentence to: “Antibiotic treatments, often topical, are sed as first line therapy for impetigo”. 

Line #19 – Line #20: Consider rewarding the sentence to: the efficacy of potential new antimicrobial compounds is first tested in vitro studies followed by in vivo studies using animal models and/or humans. 

Line #20: Consider changing the term “tool” to “means” and the term “screening” vs. “examining” or “investigating” or “exploring”.  Screening does not flow well.

Line #39 – Adverse effects has already been abbreviated (Line #25), therefore use the abbreviation.  May consider abbreviating gastrointestinal (GI) as well since it is repeated multiple times later….

INTRODUCTION

Line #31: Consider changing “disease” to “Infection”.

Line #39 - Line #48: You mention impetigo presents in two forms; bullous and non-bullous.  Then you go on to define non-bullous and then bullous, and then again non-bullous.  Consider discussing the bullous type first and then non-bullous since you discuss this one more in depth.

Line #49: The term “treatment” is repeated three time in this short sentence.  Consider just making it simple like: “First-line therapies for impetigo treatment are often topical agents for faster symptom resolution”.

Line#53: I don’t think the term “severely” is need here, consider deleting.

Line #55 and Line #56: Not sure if “urgent” and “critical” are both needed.  May just say critical

Line #67: Consider deleting “programs”, not needed, just end with drug development.

Line #75: The technical complexity does not read and flow well.  Consider rewarding to: “the technical complexity of conducting and interpreting of the study”, and the …..

Line #77 – Line #78: Revise this first sentence to read and flow better: “The selection of animal models for the purpose of examination and evaluation of potential antibacterial agents, the investigators have to be able to create the infection by the target human pathogen.  (not sure if I like my suggestion, but I tried).

Line #84: Change “by” to “from”.

Line #93 – Line#95: A little repetitive words of impetigo and animal models.  Consider simplifying to: Here we critically review the animal models considered for selection of drug candidates, discuss the methodology, strengths, limitations, and the rationale for their application. 

Delete the last sentence and the next title, “impetigo animal models.  Finish this section (line #92-Line#99) with: “a summary of various animal models is presented in Table 1, and Figure 1; in addition, the strengths and the limitations of the studies is summarized in Table 2.

Syrian Hamster Impetigo Model

Line #119: Just say: were examined photographically and the effect….

Line #122: Delete “were”, the lesions then ruptured; or did someone had to rupture the lesions?

Liine#165-Line#167: Consider rewarding this section to: The hamster model has limited value for the experimental purposes in the in vivo studies of topical antimicrobials due to a substantial difference between human and hamster immune system. 

Mouse Impetigo Model

Line #182: Delete “of cost” just end with lose cost.

Line #186 – Line#187: I think if you consistently use “skin disorder” or skin infection may be more appropriate.  Skin disease sounds a little hard.  Just a thought.

Line #188: Delete “firstly”, not needed nor adds to the content.

Line #192, Line #193, and multiple other lines:  The cyclophosphamide original abbreviation is “CY”; then “Cy” is used in other sections.  Please consider correcting.

Humanized Mouse Model

Line #254: Delete “of the” and just say, “the most reproduceable method tested to achieve superficial skin damage”. 

Line#262 – Line#264: This is a minor comment: A number of important limitations exists for hu-mouse model.  It is technically complex, requires a source of human tissue, and surgical and post-surgical animal nursing skills to successfully achieve the graft. 

Line #265: You used “prevented” past tense, so use “removed”, past tense as well.

Line #278: the words “limitation” and “limit”.   Consider using restriction for the first on< the identifiable models have a number of restrictions that limit……

Mouse Skin Tape Stripping

Line #290: Delete “tape” no need since it has been mentioned.

Line #293: Just needs few for separation of “The mice (Pleural) were killed, wounds were excised, processed, and cultured to …….

Line #324: End the sentence with reliable and reproduceable mode. 

Line#330:  MIC and MBC are listed for the first time.  Although they are common abbreviations, you may consider defining first, i.e., minimum inhibitory concertation and minimum bactericidal concertation.

Line #402: Why is the mouse suture-superficial…..italicized?  

Line #418: Revise this section to: Given the strengths and limitations of models reviewed, when planning for the in vivo efficacy of a potential topical antimicrobial therapies, key considerations beyond their practical feasibility in a given laboratory should include:  List 1, 2, 3, 4.

Tables:

Nice, no comments

Figure(s):

Nice and easy to follow, no comments.

References:

Did not get a chance to

Author Response

Reviewer #4

GENERAL COMMENT

[Comment*] Very interesting review enjoyed reading the manuscript.  It is well written and flows well.  I just have few minor comments for the authors for clarification and/or adjustment if they agree to do so. 

[#Response#]: We are thankful to reviewer #1 for the kind words and positive comments.

Minor comments

ABSTRACT

  1. [Comment*] Line #17 - Line #18: The term “treatment” is repeated three times, consider rewarding the sentence to: “Antibiotic treatments, often topical, are sed as first line therapy for impetigo”. 

[#Response#]: Thanks, and it is now modified as “Antibiotic treatments, often topical, are used as the first-line therapy for impetigo”. (Please see Page #1 and Line #17–18)

  1. [Comment*] Line #19 – Line #20: Consider rewarding the sentence to: the efficacy of potential new antimicrobial compounds is first tested invitro studies followed by in vivo studies using animal models and/or humans.

[#Response#]: Thanks, and it is now modified as suggested “The efficacy of potential new antimicrobial compounds is first tested in in vitro studies and if effective, followed by in vivo studies using animal models and/or humans”. (Please see Page #1 and Line #18–20)

  1. [Comment*] Line #20: Consider changing the term “tool” to “means” and the term “screening” vs. “examining” or “investigating” or “exploring”.  Screening does not flow well.

[#Response#]: Thanks, we have now changed the terms “tool” to “means” and “screening” to “investigating”. (Please see Page #1 and Line #20)

  1. [Comment*] Line #39– Adverse effects has already been abbreviated (Line #25), therefore use the abbreviation.  May consider abbreviating gastrointestinal (GI) as well since it is repeated multiple times later….

[#Response#]: Thanks, and we do not think we mentioned ‘Adverse effects’ and ‘gastrointestinal (GI)’ in the stated line number. (Please see Page #1 and Line #25 and Page #2 and Line #35–40)

INTRODUCTION

  1. [Comment*] Line #31: Consider changing “disease” to “Infection”.

[#Response#]: Thanks, and the word "disease” is now replaced with “infection”. (Please see Page #1 and Line #31)

  1. [Comment*] Line #39 - Line #48: You mention impetigo presents in two forms; bullous and non-bullous.  Then you go on to define non-bullous and then bullous, and then again non-bullous.  Consider discussing the bullous type first and then non-bullous since you discuss this one more in depth.

[#Response#]: Thanks, and the paragraph is now modified as “Impetigo presents in two forms – bullous and non-bullous [4]. Bullous impetigo is generally caused by S. aureus and clinically presents with large fragile, flaccid, fluid filled bullae (up to 2 cm in diameter) that less readily rupture into thin and brown crusts [4, 7]. Non-bullous or, crusted impetigo, is the most common form of impetigo caused by both S. aureus and GAS [7]. It begins with a vesicle located on an erythematous base, which is easily ruptured resulting in superficial ulceration covered with purulent discharge that then dries as an adhering yellow crust.” (Please see Page #1–2 and Line #38–44)

  1. [Comment*] Line #49: The term “treatment” is repeated three time in this short sentence.  Consider just making it simple like: “First-line therapies for impetigo treatment are often topical agents for faster symptom resolution”.

[#Response#]: Thanks, and it is now modified as “Antibiotics, often topical, are used as the first-line treatments for impetigo for faster symptom resolution”. (Please see Page #2 and Line #49–50)

  1. [Comment*] Line#53: I don’t think the term “severely” is need here, consider deleting.

[#Response#]: Thanks, and “severely” is now deleted. (Please see Page #2 and Line #53)

  1. [Comment*] Line #55 and Line #56: Not sure if “urgent” and “critical” are both needed.  May just say critical

[#Response#]: Thanks, and the word “urgent” is now deleted and “critical” is kept as suggested. (Please see Page #2 and Line #55)

  1. [Comment*] Line #67: Consider deleting “programs”, not needed, just end with drug development.

[#Response#]: Thanks, and the word “program” is now deleted. (Please see Page #2 and Line #67)

  1. [Comment*] Line #75: The technical complexity does not read and flow well.  Consider rewarding to: “the technical complexity of conducting and interpreting of the study”, and the …..

[#Response#]: Thanks, and this part is now modified as “…the technical complexity of conducting and interpreting of the study, and the ethical acceptability of the animal used in the study [33, 34].”. (Please see Page #2 and Line #74–76)

  1. [Comment*] Line #77 – Line #78:Revise this first sentence to read and flow better: “The selection of animal models for the purpose of examination and evaluation of potential antibacterial agents, the investigators have to be able to create the infection by the target human pathogen.  (not sure if I like my suggestion, but I tried).

[#Response#]: Thanks, and this part is now modified as “For animal models aimed at screening and evaluating potential antibacterial agents for human use, the models should be able to establish infections of the target human pathogens [34, 35].”. (Please see Page #2 and Line #77–78)

  1. [Comment*] Line #84: Change “by” to “from”.

[#Response#]: Thanks, and the word “by” is now replaced with the word “from”. (Please see Page #2 and Line #84)

  1. [Comment*] Line #93 – Line#95:A little repetitive words of impetigo and animal models.  Consider simplifying to: Here we critically review the animal models considered for selection of drug candidates, discuss the methodology, strengths, limitations, and the rationale for their application. Delete the last sentence and the next title, “impetigo animal models.  Finish this section (line #92-Line#99) with: “a summary of various animal models is presented in Table 1, and Figure 1; in addition, the strengths and the limitations of the studies is summarized in Table 2.

[#Response#]: Thanks, and this part is now modified as “Here, we critically review the animal models investigated to mimic clinical impetigo and discuss their methodology, strengths, limitations, and the rationale for their application in evaluating impetigo drug candidates. A summary of the various animal models is presented in Table 1 and Figure 1, and the strengths and limitations of the models are also summarized in Table 2.”. (Please see Page #3 and Line #93–97)

Syrian Hamster Impetigo Model

  1. [Comment*] Line #119:Just say: were examined photographically and the effect….

[#Response#]: Thanks, and this part is now modified as “The appearances of the lesions were examined photographically and the effect…”. (Please see Page #3 and Line #113–114)

  1. [Comment*] Line #122:Delete “were”, the lesions then ruptured; or did someone had to rupture the lesions?

[#Response#]: Thanks, and the word “were” is now deleted. (Please see Page #3 and Line #117)

  1. [Comment*] Liine#165-Line#167:Consider rewarding this section to: The hamster model has limited value for the experimental purposes in the in vivo studies of topical antimicrobials due to a substantial difference between human and hamster immune system. 

[#Response#]: Thanks, and this part is now modified as “The hamster model was designed almost four decades ago, and currently little data on the response of this model to currently marketed topical antibiotics is available [57]. In addition, the use of hamsters for experimental purposes in the investigation of immune responses to skin infections has now been diminished due to a substantial difference between human and hamster immune system [58].

In sum, this model appears to have limited value in evaluating in vivo efficacy of potential topical antibacterial agents targeting impetigo and other SSTIs.” (Please see Page #4 and Line #160–166)

Mouse Impetigo Model

  1. [Comment*] Line #182:Delete “of cost” just end with lose cost.

[#Response#]: Thanks, and the phrase “of use” is now deleted. (Please see Page #4 and Line #171)

  1. [Comment*] Line #186 – Line#187:I think if you consistently use “skin disorder” or skin infection may be more appropriate.  Skin disease sounds a little hard.  Just a thought.

[#Response#]: Kindly noted and we have now replaced “skin diseases” with “skin infections” in this part. (Please see Page #4 and Line #177)

  1. [Comment*] Line #188:Delete “firstly”, not needed nor adds to the content.

[#Response#]: Thanks, and the word “firstly” is now deleted. (Please see Page #4 and Line #178)

  1. [Comment*] Line #192, Line #193, and multiple other lines:  The cyclophosphamide original abbreviation is “CY”; then “Cy” is used in other sections.  Please consider correcting.

[#Response#]: Thanks, and we have now modified the original abbreviation as “Cy”. (Please see Page #4 and Line #178)

Humanized Mouse Model

  1. [Comment*] Line #254: Delete “of the” and just say, “the most reproduceable method tested to achieve superficial skin damage”. 

[#Response#]: Thanks, and we have now modified this part as “…the most reproducible method tested to achieve superficial skin damage”. (Please see Page #5 and Line #243–244)

  1. [Comment*] Line#262 – Line#264:This is a minor comment: A number of important limitations exists for hu-mouse model.  It is technically complex, requires a source of human tissue, and surgical and post-surgical animal nursing skills to successfully achieve the graft. 

[#Response#]: Thanks, and we have now modified this part as “The hu-mouse model is associated with a number of important limitations. It is technically complex, requiring a source of human tissue and surgical and post-surgical animal nursing skills to successfully achieve the grafts [30, 67].” (Please see Page #6 and Line #251–253)

  1. [Comment*] Line #265:You used “prevented” past tense, so use “removed”, past tense as well.

[#Response#]: Thanks, and we have now changed the word “removes” to “removed”. (Please see Page #6 and Line #254)

  1. [Comment*] Line #278:the words “limitation” and “limit”.   Consider using restriction for the first on< the identifiable models have a number of restrictions that limit……

[#Response#]: Thanks, and we have now modified this part as “In addition, the identified models have a number of limitations that restrict their feasibility in evaluating the clinical efficacy of topical antibacterial agents”. (Please see Page #6 and Line #266–267)

Mouse Skin Tape Stripping

  1. [Comment*] Line #290: Delete “tape” no need since it has been mentioned.

[#Response#]: Thanks, and the word “tape” is now deleted. (Please see Page #6 and Line #278)

  1. [Comment*] Line #293: Just needs few for separation of “The mice(Pleural) were killed, wounds were excised, processed, and cultured to …….

[#Response#]: Thanks, and we have now modified this part as “Eighteen hours after the last topical treatment, the mice were killed, the wounds (about 2 cm2) were excised, processed, and cultured to determine the number of viable bacteria per unit area of skin.”. (Please see Page #6 and Line #280–282)

  1. [Comment*] Line #324:End the sentence with reliable and reproduceable mode. 

[#Response#]: Thanks, and we have now modified this part as “From these inconsistent findings, it is understood that the mouse skin tape-stripping is unlikely to be a reliable and reproducible model for impetigo.” (Please see Page #7 and Line #310–312)

  1. [Comment*] Line#330: MIC and MBC are listed for the first time.  Although they are common abbreviations, you may consider defining first, i.e., and minimum bactericidal concertation minimum inhibitory concertation.

[#Response#]: Thanks, and we have now spelled out the abbreviations as “(minimum inhibitory concentration and minimum bactericidal concentration).” (Please see Page #7 and Line #318)

  1. [Comment*] Line #402:Why is the mouse suture-superficial…..italicized?  

[#Response#]: Thanks, and we have now unitalicized it. (Please see Page #9 and Line #402)

  1. [Comment*] Line #418: Revise this section to: Given the strengths and limitations of models reviewed, when planning for the in vivoefficacy of a potential topical antimicrobial therapies, key considerations beyond their practical feasibility in a given laboratory should include:  List 1, 2, 3, 4.

[#Response#]: Thanks, and we have now modified this part as “Given the strengths and limitations of models reviewed, when planning for the in vivo efficacy evaluation of potential topical antibacterial therapies, key considerations beyond their practical feasibility in a given laboratory should include: 1) the establishment of reproducible, controllable infections with measurable endpoints of efficacy, 2) reliable and simple quantification of pre-and post-treatment viable bacteria counts, and 3) correlation of in vivo results with clinical effectiveness in humans.” (Please see Page #15 and Line #426–431)